# Bridging the Gap: Endothelial Dysfunction and the Role of iPSC-Derived Endothelial Cells in Disease Modeling

**DOI:** 10.3390/ijms252413275

**Published:** 2024-12-11

**Authors:** Chiara Sgromo, Alessia Cucci, Giorgia Venturin, Antonia Follenzi, Cristina Olgasi

**Affiliations:** 1Department of Health Sciences, School of Medicine, University of Piemonte Orientale, 28100 Novara, Italy; chiara.sgromo@uniupo.it (C.S.); alessia.cucci@uniupo.it (A.C.); giorgia.venturin@uniupo.it (G.V.); 2Department of Translational Medicine, School of Medicine, University of Piemonte Orientale, 28100 Novara, Italy; cristina.olgasi@med.uniupo.it

**Keywords:** endothelial dysfunction, induced pluripotent stem cells, endothelial cells, diabetes, cardiovascular diseases, neurodegenerative disorders

## Abstract

Endothelial cells (ECs) are crucial for vascular health, regulating blood flow, nutrient exchange, and modulating immune responses and inflammation. The impairment of these processes causes the endothelial dysfunction (ED) characterized by oxidative stress, inflammation, vascular permeability, and extracellular matrix remodeling. While primary ECs have been widely used to study ED in vitro, their limitations—such as short lifespan and donor variability—pose challenges. In this context, induced iECs derived from induced pluripotent stem cells offer an innovative solution, providing an unlimited source of ECs to explore disease-specific features of ED. Recent advancements in 3D models and microfluidic systems have enhanced the physiological relevance of iEC-based models by better mimicking the vascular microenvironment. These innovations bridge the gap between understanding ED mechanisms and drug developing and screening to prevent or treat ED. This review highlights the current state of iEC technology as a model to study ED in vascular and non-vascular disorders, including diabetes, cardiovascular, and neurodegenerative diseases.

## 1. Introduction

The endothelium is a dynamic layer of cells lining the inner surfaces of blood and lymphatic vessels, essential in maintaining vascular integrity [1] and in regulating several physiological mechanisms, such as blood flow, vascular tone, nutrient exchange, and the passage of molecules and cells between the bloodstream and the surrounding tissues [2].

The endothelial cells (ECs) mediate all these functions, and moreover, they are actively involved in the regulation of hemostasis and thrombosis, in wound healing, and in coordinating both innate and adaptive immunity reactions [3,4,5].

Given their multiple functions, an impairment in ECs integrity and/or activity causes the endothelial dysfunction (ED), which is closely linked to the development of cardiovascular diseases (CVD) but also of metabolic and neurodegenerative diseases (ND) [6,7,8]. ED is marked by a loss of normal endothelial function, including impaired vasodilation, increased vascular permeability, and excessive pro-inflammatory signaling. These alterations are driven by a complex interplay of factors such as oxidative stress, altered shear stress, tight junction disruption, and changes in the extracellular matrix (ECM) [9].

Over the years, in vitro modeling of the ED has been conducted on primary ECs, isolated from human or animal tissues. Although useful, these in vitro systems suffer from short cell lifespans, variability between donors, and difficulties in scaling up for large studies. The advent of induced pluripotent stem cell (iPSC)-derived endothelial cells (iECs) has revolutionized this field, offering a renewable, patient-specific source of ECs that can be used to model diseases and to study the underlying mechanisms of ED. iECs allow researchers to investigate how specific genetic mutations or environmental factors impact the endothelial differentiation and function, helping the investigation of the molecular pathways involved and providing a platform for testing potential therapies.

In this review, we aim to summarize the mechanisms involved in the ED and the uses of iECs as models for several pathological conditions such as CVD, diabetes, and ND.

## 2. Endothelial Dysfunction

ED is characterized by impaired vascular tone regulation, blood flow maintenance, and inflammation prevention. Several factors, such as an increase in oxidative stress, ECM remodeling, and hypertension, can contribute to the onset of ED, making it a multifactorial phenomenon [10]. Indeed, the ED typically begins with early events such as endothelial activation and inflammation, followed by oxidative stress and apoptosis. However, the exact order of these events can vary depending on the primary cause, highlighting the ED dynamic progression [9].

### 2.1. Role of Oxidative Stress in Endothelial Function

The normal structure and function of blood vessels is maintained by balancing vasodilation, mediated by endothelium-derived relaxing factors (EDRFs), and vasoconstriction, controlled by endothelium-derived constricting factors (EDCFs) [11] (Figure 1A). ECs release a variety of vasodilators such as nitric oxide (NO), prostacyclin (PGI2), and C-type natriuretic peptide, along with vasoconstrictors like endothelin, angiotensin II, prostaglandin H_2_, and thromboxane A2 [12,13,14]. An imbalance in the secretion of these vasoactive substances can lead to increased production of reactive oxygen species (ROS), contributing to ED. ROS are products of cellular metabolism, primarily formed as reactive intermediates of oxygen in ECs [15]. Oxidative stress occurs when ROS production surpasses the antioxidant defense capabilities of the cells or when antioxidant enzymes are compromised [16,17]. One key effect of ROS-induced ED is a reduction in NO bioavailability, which promotes vasoconstriction and thrombosis, the release of pro-inflammatory cytokines, and the proliferation of neighboring cells [18]. Clinically, the release of NO from ECs can be assessed through changes in flow-mediated dilation (FMD), a measure that strongly correlates with endothelial damage [19]. Furthermore, disruptions in NO production or signaling are associated with various cardio-metabolic conditions [20]. NO is produced by three isoforms of nitric oxide synthase (NOS): neuronal NOS (nNOS) in the nervous system, inducible NOS (iNOS) during inflammation by macrophages, ECs, and smooth muscle cells; and endothelial NOS (eNOS) in vascular ECs, cardiomyocytes, and platelets, where it regulates vascular health and homeostasis [21]. Reduced NO availability is due to (1) inactivation of NO through its reaction with superoxide (O_2_^−^) to form peroxynitrite (ONOO^−^), diminishing NO’s effective concentration; (2) decreased NO production due to reduced eNOS expression; (3) insufficient eNOS substrates or cofactors; and (4) alterations in eNOS activity, such as uncoupling [22]. Oxidative stress and elevated ROS levels also promote protein oxidation, which triggers the upregulation of pro-inflammatory mediators like interleukin 1 beta (IL-1β), interleukin 6 (IL-6), tumor necrosis factor alpha (TNF-α), and adhesion molecules such as vascular cell adhesion molecule 1 (VCAM-1) and intercellular adhesion molecule 1 (ICAM-1) in ECs [23,24]. These pro-inflammatory cytokines contribute to vascular insulin resistance and the infiltration of monocytes into the endothelium, leading to chronic inflammation and further endothelial injury [25]. Additionally, ROS can activate the nuclear factor kappa-light-chain-enhancer of activated B cells (NF-κB) signaling pathway, which translocates to the nucleus to induce the expression of genes like *VCAM-1*, *ICAM-1*, *IL-6*, and *TNF-α*, ultimately resulting in ED [26] (Figure 1A). In conclusion, the imbalance of oxidative processes within the vasculature plays a pivotal role in the initiation and progression of ED, which can lead to the vascular clinical outcomes of CVD and diabetes.

### 2.2. Role of Shear Stress, Tight Junctions, and Permeability in Endothelial Function

Shear stress is fundamental for the maintenance of vascular homeostasis since ECs act as sensors of shear stress through the activation of the integrin signaling that induces several downstream pathways involved in cell adhesion, migration, and angiogenesis. Therefore, changing the shear stress can result in the activation of different receptors or in the inhibition of integrins, tyrosine kinase receptors, caveolins (Cav), G proteins, ion channels, and adhesion receptors [27]. Under laminar shear stress (LSS), anti-inflammatory, anti-coagulant, antioxidant, and anti-apoptotic effects are induced since LSS prevents monocyte adhesion and vascular smooth muscle cell proliferation, induces eNOS expression, and activates factor E2-related factor 2 (*Nrf2*) and Krüppel-like factor 2 (*KLF2*) [28].

On the contrary, the exposure to low (time-average: 10–12 dyne/cm^2^) or oscillatory shear stress (time-average: close to 0) promotes coagulation, oxidation, and apoptosis and increases EC permeability, resulting in altered EC function [29] (Figure 1B). These effects are mediated by mechanotransduction pathways activated through mechanosensors that damage glycocalyx integrity, cytoskeleton arrangement, and cell–cell junctions [28,29,30,31,32]. The glycocalyx, comprising glycoproteins and proteoglycans, such as heparan sulfate and glypican-1, responds to hemodynamic forces, producing NO and thus maintaining a low permeability in LSS [33]. In low or oscillatory shear stress, heparan sulfate is degraded, resulting in increased endothelial permeability and inflammation [34]. Some transcription factors can also be activated, such as nuclear factor kappa-light-chain-enhancer of activated B cells (NF-κB), activator protein-1 (AP-1), Yes1-associated transcriptional regulator (YAP)/Tafazzin (TAZ)/TEA domain transcription factor 1 (TEAD), and hypoxia-inducible factor 1α (HIF-1α), leading to increased inflammatory signaling [35]. The induced inflammation can increase the release of vasoactive agents and inflammatory mediators, including VEGF, histamine, thrombin, IL-1β, and TNF-α, disrupting cell–cell junctions and leading to hyperpermeability and edema (Figure 1B) [36]. Finally, lipopolysaccharide (LPS), a highly pathogenic endotoxin responsible for organ dysfunction in sepsis bacterial endotoxin, exacerbates this process by activating RhoA, ROCK, Rho kinase, and other guanosine triphosphate (GTPases), which destabilize cytoskeletal and junctional integrity [37].

### 2.3. Role of Extracellular Matrix in Endothelial Dysfunction

The ECM provides structural support, modulates the cytoskeletal organization, and influences EC behavior through biochemical and mechanical signals. In healthy blood vessels, the ECM facilitates proper EC adhesion, migration, and proliferation, influencing several processes such as vessel formation and wound healing. The key components of ECM are collagens, proteoglycans (PGs), glycosaminoglycans (GAGs), elastin, laminins, fibronectin, and syndecans, forming, all together, a three-dimensional network of molecules that support the structure of cells and tissues [38]. Moreover, cells connect to ECM and surrounding cells through adhesion molecules linked to the actin cytoskeleton. Therefore, the interplay between ECM and cytoskeleton is essential [39], and a destabilization of ECM, direct or related to altered cytoskeletal proteins, can induce EC dysfunction (Figure 1C). For instance, the cytoskeletal protein vinculin binds to other cytoskeletal proteins such as actin, α-actinin, talin, and vinexin to secure the cytoskeleton to ECM at sites of adherens junctions and focal adhesions [40]. Specifically, in ECs, vinculin is found at β-catenin and VE-cadherin junctions, maintaining the integrity of the endothelial barrier by regulating vascular permeability and by preventing immune cells and molecules from leaking [41]. Recent studies reveal that low or oscillatory shear stress inactivates vinculin in murine ECs, leading to endothelial injury. This effect occurs as flow disruption activates G protein-coupled receptor kinase 2 (GRK2), which moves to cell adhesion sites and phosphorylates vinculin, destabilizing the cytoskeleton (Figure 1C) [42]. GRK2 inhibition reactivates vinculin, reducing inflammation, but the mechanism behind GRK2 activation needs to be elucidated [43]. Another protein involved in EC function is developmental endothelial locus-1 (DEL-1), a glycoprotein secreted by ECs into the ECM to prevent immune cell adhesion and infiltration [44]. DEL-1 binds to the lymphocyte function-associated antigen 1 (LFA-1) integrin on immune cells, blocking its interaction with endothelial ICAM-1, which limits leukocyte adhesion and extravasation across the endothelial barrier, restricting immune cell recruitment [45]. DEL-1 also regulates the activity of matrix metalloproteinases (MMPs), especially MMP-2, which can degrade tight junction proteins like occludin, weakening the endothelial barrier and increasing permeability. DEL-1 inhibits MMP-2 by binding to αvβ3 integrin on ECs and immune cells, preventing αvβ3 integrin from activating pro-MMP-2, which helps preserve the endothelial barrier [46]. Overexpression of DEL-1 in primary ECs showed protective effects against ED and fibrosis, suggesting that DEL-1 and similar ECM proteins could offer therapeutic targets to prevent ED and vascular inflammation under pathological conditions [46].

## 3. Primary Endothelial Cells and Their Limitations

Mechanistic in vitro studies on ED serve as an early predictor of CVD, including atherosclerosis. The identification of biomarkers associated with ED offers critical insight into the early stages of these diseases, enabling timely diagnosis and intervention. To offer an accurate study of the molecular pathways determining ED and its related pathologies, in vitro culture of primary ECs has been used to study vascular regeneration, tissue engineering, and drug development. Due to the challenges in assessing in vivo the human vascular endothelial tissue, cell culture technologies have been developed to replicate the dysfunctional vascular wall alterations. These models serve as important tools for investigating EC physiology and elucidating the mechanisms of EC interaction with other cells and different mediators [47]. Human umbilical vein endothelial cells (HUVECs) have been widely used as an endothelial model allowing the study of key mechanisms involved in ED, such as NO production, inflammation, and oxidative stress, and to test therapeutic agents aimed at restoring endothelial function [48,49].

Besides HUVECs, circulating ECs derived from peripheral or cord blood have been considered. Recently, two different cell populations derived from mononuclear cells (MNCs) have been recognized as early and late endothelial progenitor cells (EPCs). Although “early EPCs” show similar features of ECs (e.g., CD31^low^, VEGFR2^low^), they have been recognized as bone marrow–derived myeloid angiogenic cells (MACs) that express typical monocyte markers (e.g., CD11b, CD14). Several studies pointed out that MACs did not form new blood vessels on their own but facilitated their establishment by releasing factors that promote angiogenesis, and so they cannot be considered as mature ECs [50]. On the other hand, by maintaining MNCs in culture in specific growth conditions, it is possible to obtain a cell population defined as “late” outgrowth EPCs or endothelial colony-forming cells (ECFCs), previously called blood outgrowth endothelial cells (BOECs), which appear as colonies of mature ECs [51].

ECFCs have been obtained from patients affected by several diseases, such as diabetes. Diabetic ECFCs displayed an altered functionality with a reduced tubulogenic capacity and migration potential compared to ECFCs from healthy controls [52]. Interestingly, a hyperglycemic environment, mimicked by high glucose in the culture medium, affects healthy ECFCs, supporting the idea that diabetes-related hyperglycemia is one of the main causes of ECFC dysfunction [53]. Moreover, ECFCs have been used not only to replicate the pathophysiology of diabetes but also in regenerative medicine therapies for diabetes [54] and in pre-clinical studies of hematological diseases, such as hemophilia A (HA) [55].

Recently, factor VIII (FVIII) has been shown to be an important player in the maintenance of ECs permeability, migration, and tubulogenesis, besides its function in hemostasis. Indeed, FVIII absence impaired HA ECFCs, but its reintroduction restored the altered migration, tubulogenesis, and permeability. In particular, FVIII promoted the activation of the focal adhesion kinase (FAK)/Src signaling pathway leading to the regulation of ECM-related genes such as nidogen 2 [56].

Despite their usefulness, the culture of primary ECs is limited by donor availability, limited replicative capacity, and challenges in obtaining sufficient cells, particularly from diseased tissues (Table 1) [57]. By contrast, iPSCs are easily obtained and undergo unlimited self-renewal, being an attractive source for generating ECs, overcoming primary cell concerns. However, iPSC-derived ECs (iECs) have some limitations as well, since they often display immature or heterogeneous phenotypes and require extensive validation to ensure functionality and stability, particularly for in vivo applications (Table 1).

## 4. iPSC-Derived Endothelial Cells

The discovery of iPSCs has revolutionized regenerative medicine thanks to their self-renewal potential, growth, and differentiation capacity that resemble the ones of embryonic stem cells. iPSCs provide an attractive, patient-specific cell source to be differentiated into virtually any cell type, including ECs. iPSCs have been generated by reprogramming several types of adult cells by introducing different combinations of pluripotency genes with a plethora of delivery systems and, since their discovery, have been differentiated into a wide range of cells [68]. However, the general approach for iPSCs differentiation is based on the knowledge coming from the developmental process characteristic of embryogenesis, which is possible to mimic in vitro to achieve the desired cell fate.

### 4.1. Endothelial Cell Types

ECs, despite sharing core functions, exhibit remarkable diversity in both origin and morphology. They can be classified based on their developmental origin, such as arterial, venous, or lymphatic, or by their morphological features as continuous, fenestrated, or discontinuous [69]. Anatomically, the vascular wall is characterized by the presence of pericytes binding ECs through the communicating junctions, which may present different levels of adhesion [70], and smooth muscle cells that surround ECs, contributing to the regulation of vascular tone, functions, and blood flow [71]. The knowledge of EC distinctions is essential for comprehending their roles in various physiological processes and developing new protocols to properly differentiate these subtypes from iPSCs.

### 4.2. Endothelial Specification of iPSCs

Each endothelial subtype has its unique set of markers and functions depending on its origin and location. However, across all subtypes, certain endothelial-specific markers like CD31, von Willebrand factor, CD144, and VEGF receptor 2 (VEGFR2) are widely recognized. The differentiation of iPSCs into venous, arterial, and lymphatic ECs is governed by the precise modulation of key signaling pathways. Generally, iPSCs differentiation starts with embryoid bodies (EBs) induction to recapitulate the gastrulation, followed by the promotion of one of the three layers, depending on the desired cell type. An essential and common step for EC differentiation is the mesoderm induction, which can be achieved through the combined action of bone morphogenetic protein 4 (BMP4), fibroblast growth factor 2 (FGF2), and a glycogen synthase kinase-3 (GSK3) inhibitor (CHIR99021); BMP4 directs iPSCs toward the mesodermal lineage in a concentration-dependent manner [72]; FGF2 supports mesoderm specification by promoting VEGFR2 expression [73]; CHIR99021 leads to the activation of transcription factors critical for mesoderm formation and significantly reduces pluripotency markers such as Octamer-binding Transcription Factor 4 (OCT4) and SRY-Box Transcription Factor 2 (SOX2) (Figure 2) [74]. Another key point for EC differentiation is the induction of VEGF signaling to promote the commitment of mesodermal cells towards the endothelial fate by activating p38-MAPK, which also regulates ETS variant transcription factor 2 (ETV2) expression [75]. Both arterial and venous EC differentiation relies on balanced VEGF signaling through VEGFR2. Indeed, several studies demonstrated that the modulation of VEGF pathways and the use of specific small molecules direct endothelial differentiation into arterial phenotypes. High VEGF levels (50 ng/mL) with cyclic adenosine monophosphate (cAMP) enhance arterial markers while suppressing venous markers like Ephrin type-B receptor 4 (EphB4) and COUP transcription factor II (COUP-TFII) (Figure 2). Notch signaling further supports arterial specification by inducing Gap Junction Protein Alpha 4 (GJA4) and p21, which inhibit the EC cycle and promote arterial identity [76,77]. On the contrary, the modulation of COUP-TFII expression, influenced by low VEGF concentrations (10 ng/mL), suppresses Notch signaling while promoting the upregulation of Ephrin receptor B4 (EphB4), thereby driving venous endothelial cell specification and differentiation by inhibiting mitotic division through p42 and p44 signaling pathways (Figure 2) [78]. Differently, lymphatic capillaries are characterized by a continuous EC layer and the absence of vascular smooth muscle cells and pericytes, which are crucial for blood capillary integrity [79]. This structural difference is due to lower expression of ECM-regulating genes, leading to an incomplete basement membrane [79]. Lymphatic endothelial cells (LECs) in humans primarily originate from embryonic veins, though organ-specific variations in lymphatic development suggest distinct cellular origins or pathways [80]. Nevertheless, the ERK/Sox18/Prox1 pathway regulates lymphatic endothelial cell differentiation [81], with ERK activation driving Sox18 expression and subsequently inducing Prox1 upregulation; COUP-TFII supports this process by enhancing lymphatic vessel endothelial hyaluronan receptor 1 (LYVE1) and Prox1 expression, enabling the transition of venous endothelial progenitors into the lymphatic lineage [82,83]. Lee et al. compared three culture conditions for obtaining LECs from iPSCs [84]. With the first condition, authors generated and cultured EBs in suspension for 30 days, with LEC markers (Prox1, LYVE1, VEGFR3, and Podoplanin) beginning to be expressed by day 7 and peaking at day 20. In the second condition, EBs were co-cultured with a murine macrophage-derived embryonic stem cell line, OP9 cells, and supplemented with VEGF-A, VEGF-C, and EGF for 30 days. Finally, the last method was based on a feeder-free culture with gelatin in the presence of the same growth factors. The co-culture system with OP9 was found to be the most effective, showing an increased expression of LEC markers and less mortality. Rufaihah et al. described a shorter differentiation protocol for LECs, where 4-day EBs treated with BMP4, VEGF-A, VEGF-C, and angiopoietin-1 (Ang-1) for 10 days [85] showed similar results to those of the previous protocol with an increase in lymphatic markers starting from the fourth day with a peak on the eleventh (Figure 2). Overall, these differentiation protocols highlight the intricate processes that shape the vascular system—arterial, venous, and lymphatic. Understanding the molecular pathways regulating differentiation into EC subtypes is crucial to recreate the diseased phenotype through the direct differentiation of patient-derived iPSCs, enabling a deeper investigation into the pathophysiology of ED and its implications for diseases such as cardiovascular disorders, diabetes, and neurodegenerative conditions.

## 5. iEC Approaches to Study Endothelial Dysfunction in Pathological Conditions

The ability to generate ECs from iPSCs (iECs) has opened promising avenues for both basic research and translational applications, offering an exciting platform to study vascular biology and developing regenerative therapies. Therefore, we explore the applications of iECs as models to study ED in CVD, diabetes, and ND.

### 5.1. iEC Can Recapitulate EC Dysfunction in Cardiovascular Disease

CVDs represent a broad spectrum of disorders affecting the heart and the vasculature, including conditions such as coronary artery disease, heart failure, and stroke. CVDs remain the leading cause of mortality worldwide and are closely associated with risk factors such as hypertension, hypercholesterolemia, smoking, poor dietary habits, and physical inactivity. Several studies have highlighted the crucial role of endothelium in the onset, progression, and clinical outcomes of CVDs. Indeed, dysregulated paracrine mediators in dysfunctional endothelium can disrupt cardiomyocyte homeostasis as a consequence of the loss of NO bioavailability. Moreover, increased levels of vasoactive peptides (ET-1 and angiotensin II) can impair vasomotor control and promote hypertrophy and fibrosis [86].

iECs can be used as a model to investigate the difference in gene expression between subjects with CVDs and healthy controls and to evaluate the response of the iECs to several drug therapies. Tang et al. utilized iECs to model cadmium (Cd)-induced atherosclerosis, revealing a clear ED phenotype characterized by impaired angiogenic capabilities. Specifically, increased serum levels of Cd have been found to be associated with a higher risk of vascular diseases, with ECs being one of the main targets of Cd-induced toxicity, which can result in atherosclerosis with a mechanism that is still unknown. iECs treated with Cd showed a reduced ability to form tube-like structures and to migrate, with a significant decrease in the number and length of formed vessels compared to untreated iECs [87]. Moreover, through transcriptomic analysis it was possible to identify an upregulation of metallothionein isoforms, associated with Cd-induced cytotoxicity, and DNA-damage-inducible genes implicated in stress signaling and subsequent apoptosis, with an enrichment of pathways associated with angiogenesis and regulation of EC function. Finally, activation of p38 and ERK pathways was related to Cd-induced EC apoptosis and iEC dysfunction, revealing a molecular mechanism of endothelial toxicity induced by Cd [88]. Furthermore, Straessler et al. used iECs to investigate whether senescence and inflammation can drive the severity of CVDs. iPSCs were obtained from patients with different CVD progression and differentiated into ECs; subsequently, they were exposed to inflammatory stimuli, demonstrating an upregulation of E-selectin and ICAM1 and a downregulation of VCAM1. Furthermore, a shortening of the telomere was observed in iECs from CVD patients compared to healthy controls, indicating that in response to an inflammatory stimulation, CVD-derived iECs can progress to a senescent state [89].

Nevertheless, the number of studies addressing ED by using iECs in CVDs is very limited, and this can be attributed to the difficulty in replicating the complex factors involved in modeling in vitro both intrinsic and extrinsic factors and in obtaining proper functional characterization and differentiation of iECs [90,91]. However, the use of iECs for tissue and graft revascularization has gained traction as a promising approach for future cell therapies; for instance, it was possible to develop a functional bi-layered vascular graft using iECs [92], and engineered matrix-embedded ECs could restore dysfunctional endothelium and ischemic tissue [93].

Moreover, iECs have become fundamental in the development of 3D models, including microfluidic systems and organoids. Although they share the same goal of replicating in vitro vascular structures, these two approaches differ significantly in design and application. Microfluidic vascular systems use microfabricated devices that include pre-engineered channels necessary to guide the fluid flow and to assure cellular interactions, offering precise control over environmental conditions such as shear stress, nutrient gradients, and biochemical signals. These systems are particularly valuable for studying vascular dynamics, drug delivery, and disease progression under highly reproducible conditions. In contrast, in organoids, iECs together with supporting cells, such as pericytes and smooth muscle cells, self-assemble to form 3D blood vessels. These organoids more accurately emulate the intricate architecture and functionality of human capillaries, including lumen formation, barrier properties, and interactions with ECM. This self-assembly process allows organoids to reflect the complexity of vascular development and in vivo remodeling, making them ideal for studying angiogenesis, tissue regeneration, and pathologies such as vascular dysfunction and tumor angiogenesis [94,95]. These approaches can improve the study of CVDs since 2D cultures are insufficient to recapitulate mechanical contraction or blood flow [96]. Finally, numerous studies have also highlighted the application of iECs in modeling CVD pathophysiology, helping in the identification of molecules and mechanisms that contribute to disease progression [97,98,99]. As an example, recently, iEC from healthy donors and familial hypercholesterolemia (FH) patients were used to investigate the effects of low-density lipoprotein receptor (LDLR) dysfunction, demonstrating that defects in the LDLR are significant contributors to the pathology. Indeed, iECs derived from patients affected by FH exhibited reduced levels of mature LDLR, which is crucial for low-density lipoprotein (LDL) uptake. This deficiency is linked to elevated blood cholesterol levels, which can damage vascular cells through oxidative stress and inflammation. This heightened vulnerability, combined with elevated cholesterol levels, may accelerate ED, which is a precursor for atherosclerosis and other CVD associated with FH [100].

### 5.2. iECs Can Model EC Dysfunction in Diabetes

Diabetes is a chronic disease characterized by high glucose blood levels related to an impaired insulin quantity or function [101]. In patients with Type 1 or Type 2 diabetes, the combined effects of hyperglycemia, insulin resistance, and elevated free fatty acids impair endothelial functionality and disrupt the balance of vasodilatory and vasoconstrictive factors, significantly increasing the risk of CVD and vasculopathy, including macrovascular complications like coronary and peripheral artery disease, as well as microvascular issues such as diabetic foot ulcers and retinopathy [102]. Several works highlighted that insulin may have a direct role in regulating EC function with a mechanism that is beyond glucose metabolism, principally regulating crucial endothelial functions, such as NO production [103]. Indeed, elevated glucose levels affect different mechanisms related to endothelial functionality, in detail increasing ROS production by reducing NO availability and vasodilation, activating inflammatory pathways such as NF-κB, promoting leukocyte adhesion, and heightening vascular inflammation. Simultaneously, the MAPK pathway remains active, promoting ET-1 production and cellular proliferation, both of which contribute to vascular constriction and inflammation. The protein kinase C (PKC) pathway is also activated, further reducing NO synthesis and increasing endothelial permeability. Finally, advanced glycation end-products (AGEs) accumulate and bind to receptors for advanced glycation end-products (RAGE) receptors, triggering oxidative stress and inflammation that exacerbate endothelial damage [104,105]. Gorashi et al. generated iECs from both type 1 and 2 diabetic patients (Di-ECs) as a model to investigate their functionality [106]. Di-iECs, compared to healthy iECs, expressed inflammation-related markers (VCAM-1, ICAM-1, and P-Selectin), showed thrombo-resistant functions, had a disorganized tubulogenic potential, expressed a higher amount of intracellular ROS, exhibited a disrupted barrier function, and had an increased permeability. Interestingly, when healthy iECs were cultured in diabetic-like medium (high glucose, urea nitrogen, and TNF-α), their functionality was similar to Di-iECs, indicating the robust influence of the microenvironment. However, the transcriptomic analysis of Di-iECs revealed an impairment in genes related to ECM organization and to ED, offering new insights into the mechanisms that may contribute to diabetic vascular complications. These results demonstrate how a diabetes-like microenvironment is not sufficient to induce a diabetes-like phenotype, supporting the role of genetic predispositions, preserved during iPSC reprogramming, and epigenetic changes associated with glycemic memory, such as DNA methylation, which perpetuate altered gene expression [107,108]. Glycemic memory, resulting from the long-term effects of early hyperglycemia, contributes to vascular complications in diabetes, even after glycemic control is restored. This “hyperglycemic endothelial memory” indicates that high glucose induces molecular changes, including oxidative stress and AGE formation, leading to mitochondrial damage and a cycle of inflammation that impairs capillary function [109].

Moreover, Di-iECs displayed increased signs of senescence, marked by a decline in their proliferative capacity and functionality, impairing mitochondrial function. Furthermore, Di-iECs demonstrated a reduced angiogenic potential, as evidenced by their limited ability to form tubular structures in vitro, reflecting a decreased capacity for new blood vessel formation. Additionally, an altered protein expression profile was demonstrated with increased levels of adhesion molecules such as ICAM-1 and VCAM-1, alongside decreased NO production, indicating a pro-inflammatory shift [110]. Recently, in a 3D model of diabetic vascular organoids, a specific subpopulation of ECs exhibited elevated ROS levels and an enhanced oxidative phosphorylation activity [111]. This metabolic profile is indicative of high oxidative stress, which is a well-known contributor to cellular damage and dysfunction in diabetic tissues. Such changes suggest that these ECs are undergoing stress responses that could be considered early markers of vascular complications in diabetes. These stressed ECs also displayed early markers of aberrant angiogenesis, or irregular blood vessel formation, which is characteristic of complications in diabetic retinopathy [111]. As we continue to explore the complexities of ED in diabetes, it becomes increasingly clear that targeted strategies aimed at restoring endothelial health are essential for mitigating cardiovascular risks and improving outcomes for diabetic patients.

### 5.3. iECs Resemble EC Dysfunction in Neurodegenerative Diseases

The blood–brain barrier (BBB) is a specialized structure primarily composed of brain microvascular endothelial cells (BMECs), pericytes, and astrocytes surrounded by neurons forming a neurovascular unit (NVU). This complex multicellular functional unit governs the intricate relationship between blood and brain tissue [112]. This barrier plays a critical role in limiting the passive diffusion of molecules into the central nervous system, thereby maintaining the brain’s microenvironment. However, the BBB impairment has been associated with several ND, including Alzheimer’s disease (AD) and Parkinson’s disease (PD) [113]. In AD, the BBB becomes compromised due to chronic inflammation, oxidative stress, and amyloid-β (Aβ) deposition, which collectively impair the endothelium’s ability to regulate nutrient transport and immune cell trafficking. This ED facilitates the infiltration of neurotoxic substances and immune cells, exacerbating neuronal damage and accelerating the disease progression [114,115]. This highlights the importance of developing in vitro BBB models to better understand disease mechanisms, facilitate drug development, and assess brain permeability for novel therapeutic agents [116]. To recreate a functional NVU, it is possible to co-culture iPSC-derived BMECs (iBMECs) with astrocytes, pericytes, and neurons. Several methodologies have been employed over the years to develop reliable BBB models, all emphasizing the importance of interactions—either direct or indirect—between brain ECs and other NVU components (Figure 3) [116]. One of the most widely used methods is the transwell system, where membrane inserts are positioned above standard culture plates. In this setup, ECs are seeded on the apical side of the inserts, while astrocytes and pericytes are cultured in the lower compartments. This arrangement allows for non-contact co-culturing, or, when cells are seeded on both sides of the inserts, contact co-culturing through the membrane pores [117]. Katt et al. utilized the transwell method to investigate BBB impairment in several ND, including AD, PD, amyotrophic lateral sclerosis (ALS), and Huntington’s disease (HD). Recently, it has been shown that ED plays a key role in the progression of the last two listed diseases, in oxidative stress and inflammation, further disrupting tight junctions at the BBB, allowing the infiltration of immune cells, amplifying neuroinflammation, and compounding neuronal damage [115,118].

Human iBMECs were generated from iPSC lines, consisting of three lines from healthy individuals and eight from patients with ND. BBB function was evaluated by measuring transendothelial electrical resistance (TEER), permeability potential, and oxidative stress, demonstrating a neurodegenerative disease-associated mutation to specific BBB dysfunctions [119]. In another study, cerebral autosomal dominant arteriopathy subcortical infarcts and leukoencephalopathy (CADASIL), the most common genetic small vessel disease caused primarily by *NOTCH3* gene mutations, was investigated. This condition is characterized by recurrent strokes, cognitive decline, and vascular dementia. By generating iPSC models from CADASIL patients, the study revealed impaired BBB function, particularly in microvascular endothelial and mural cells, indicating early neurovascular dysfunction [120]. A more recent advancement in BBB modeling involves the use of organ-on-chip technologies, which offer a cutting-edge approach by simulating in vivo microenvironments. These 3D models resemble blood vessels, featuring flow channels that generate shear stress on ECs. The basic design comprises two cell culture chambers separated by a porous membrane. ECs are seeded in one chamber, while other cell types are placed in the adjacent chamber. This design allows for substance transfer and cellular interactions like those observed in the transwell systems [121]. Qosa et al. differentiated iPSCs into ECs and astrocytes obtained from ALS patients carrying *SOD1* mutations to study the role of P-glycoprotein (P-gp). The BBB reconstruction by the transwell method evidenced an upregulation of P-gp in mutated iECs. Astrocytes expressing the mutant SOD1 were shown to drive this upregulation through ROS and Nrf2 and NFκB signaling pathways. These findings underscore the contribution of astrocyte-mediated P-gp regulation to drug resistance in ALS, highlighting the necessity for targeted therapeutic strategies to overcome BBB limitations [122,123]. Another promising method for generating 3D BBB models is the spheroid technique. Spheroids are formed by aggregating brain ECs, astrocytes, and pericytes. The cells are cultured in low-adhesion plates or with scaffolding materials that prevent adherence to surfaces, encouraging the formation of spherical clusters. As spheroids grow, they develop gradients of nutrients and oxygen that mimic living tissue conditions. This model enhances the study of cellular interactions, nutrient diffusion, and drug responses, which are critical for ND research. However, challenges persist, such as variability in cell distribution within spheroids, which can complicate experimental results [124]. The hydrogel model is another innovative approach that constructs a 3D network by encapsulating NVU cells—such as ECs, astrocytes, and pericytes—within a hydrophilic polymer matrix. These biocompatible hydrogels are engineered to mimic the ECM, creating a realistic environment for BBB study. Researchers can customize the hydrogel’s mechanical and biochemical properties, controlling factors like stiffness, porosity, and growth factor release. This level of control allows for detailed examinations of BBB function, cell interactions, and drug delivery systems, making it valuable for developing treatments for ND [125]. Lastly, vascularized brain organoids represent an advanced 3D model derived from iPSCs, designed to replicate the structure and function of brain tissue, including essential vascular components for nutrient and oxygen delivery. These organoids integrate several cell types, including neurons, glial cells, and ECs, which are critical for forming blood vessel-like structures. The organoids are created using specific culture techniques that promote self-organization, often utilizing scaffolding materials to support cell growth. This innovative model is valuable for studying brain development and neural network formation, as well as for modeling ND by introducing specific mutations or environmental factors to observe disease mechanisms. The inclusion of vascular structures allows more accurate drug testing, providing insights into how therapeutic agents cross the BBB and affect brain cells. Despite their advantages in enhancing physiological relevance and facilitating the study of complex cellular interactions, developing vascularized brain organoids can be technically challenging, and variability in structure may complicate data interpretation. Overall, organoids offer significant potential in neuroscience and regenerative medicine, making them important tools for understanding brain function and developing treatments for several conditions [126]. The intricate relationship of ECs in the BBB is crucial for maintaining brain homeostasis. The advanced in vitro described models have significantly enhanced our understanding of BBB function and highlight ED as a critical nexus for understanding ND and for developing novel therapeutic strategies.

## 6. Conclusions

The advent of iECs has significantly advanced research, providing an effective model for replicating and studying the cellular and molecular features of ED. Surpassing limitations associated with primary ECs obtained through several differentiation protocols, iECs can recapitulate endothelial phenotypes, enabling investigations into disease-specific mechanisms, such as NO dysregulation, inflammatory responses, oxidative stress, and ECM regulation. Here, we summarized that it is possible to recapitulate pathophysiological conditions in vitro and analyze endothelial behavior in response to specific disease stimuli. This approach has been particularly effective in modeling the inflammatory characteristics of ED in CVD, the hyperglycemic effects in diabetes, and the compromised BBB in ND. However, several challenges remain; scalability is a major concern, as producing iECs in sufficient quantities for clinical use is costly and difficult to standardize. Reproducibility can be impacted by variability in culture and differentiation conditions, and additionally, the cost of complex models, such as organoids or microfluidic systems, limits their widespread use. Ethical concerns regarding genetic manipulation and safety also need to be addressed, especially in the context of therapeutic applications. To overcome these issues, future research must focus on developing more cost-effective production methods and improving protocol consistency by integrating 3D and microfluidic models. Combining these systems with iECs and dynamic stimuli, such as shear stress and biochemical gradients, can better mimic the physiological conditions. iECs hold the potential to revolutionize vascular research by offering more reliable and versatile models for a wide range of applications. These include accelerating drug discovery processes by providing human-relevant systems to test efficacy and toxicity, advancing personalized medicine through patient-specific disease modeling, and contributing to the development of innovative therapies for vascular disorders. By addressing existing limitations, iECs could bridge critical gaps between in vitro studies and in vivo outcomes, remodeling the future of biomedical research and clinical applications.

## Figures and Tables

**Figure 1 ijms-25-13275-f001:**
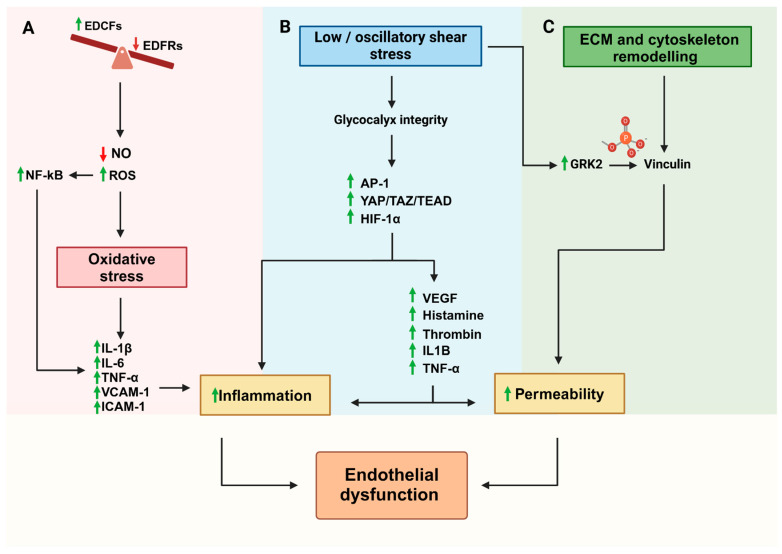
Mechanisms of endothelial dysfunction. (**A**) Oxidative stress: An imbalance between endothelium-derived constricting factors (EDCFs) and relaxing factors (EDFRs) results in reduced nitric oxide (NO) levels and increased reactive oxygen species (ROS). This activates NF-κB, promoting the expression of pro-inflammatory mediators such as IL-1β, IL-6, TNF-α, VCAM-1, and ICAM-1. (**B**) Low or oscillatory shear stress: Disruption of glycocalyx integrity activates the transcriptional regulators AP-1, YAP/TAZ/TEAD, and HIF-1α. This promotes the release of inflammatory mediators such as VEGF, histamine, thrombin, IL-1B, and TNF-α, driving inflammation and increasing endothelial permeability. (**C**) ECM and cytoskeletal remodeling: Alterations in ECM and cytoskeletal dynamics, mediated by GRK2 and vinculin, exacerbate vascular permeability. These structural changes further amplify the endothelial barrier dysfunction. Together, these processes culminate in inflammation, increased permeability, and ultimately, ED. ECM: extracellular matrix; eNOS: endothelial nitric oxide synthase; NO: nitric oxide; AP-1: activating protein 1; YAP: Yes1-associated transcriptional regulator; TAZ: Tafazzin; TEAD: TEA domain transcription factor 1; HIF-1α: hypoxia-inducible factor 1 subunit alpha; VEGF: vascular endothelial growth factor; IL-1B: interleukin beta1; TNF-α: tumor necrosis factor alpha; GRK2: G protein-coupled receptor kinase 2; and NF-Kb: nuclear factor kappa-light-chain-enhancer of activated B cells.

**Figure 2 ijms-25-13275-f002:**
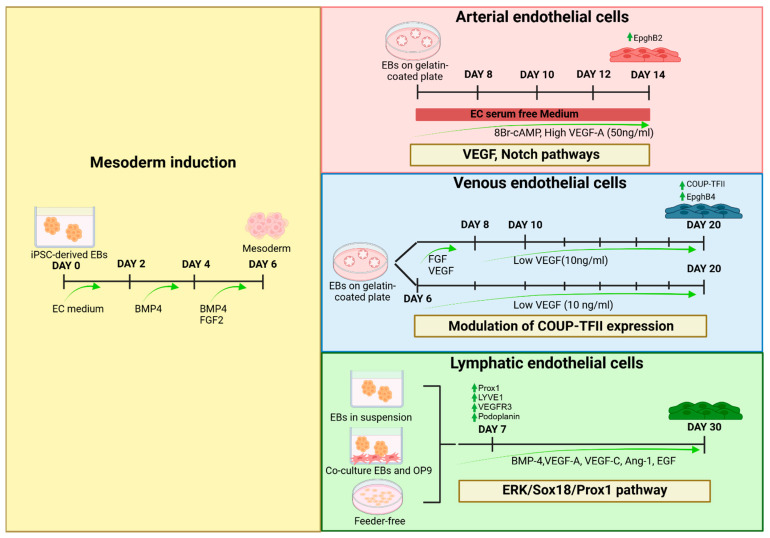
Differentiation of iPSCs into endothelial cell subtypes. Differentiation protocols for deriving arterial, venous, and lymphatic ECs from iPSCs. Mesoderm induction (yellow panel): iPSC-derived EBs are cultured in EC medium and treated sequentially with BMP4 (days 2–4) and a combination of BMP4 and FGF2 (days 4–6) to induce mesoderm specification. Arterial ECs (red panel): EBs, after mesoderm specification, are plated on gelatin-coated plates and treated with EC serum-free medium supplemented with 8Br-cAMP and a high level of VEGF-A (50 ng/mL). Activation of VEGF and Notch signaling pathways promotes arterial differentiation, marked by EphB2 expression, by day 14. Venous ECs (blue panel): After mesoderm specification, EBs are cultured with FGF and VEGF (days 6–8), followed by low VEGF (10 ng/mL) from day 10 onward. Modulation of COUP-TFII expression drives venous differentiation, as indicated by COUP-TFII and EphB4 expression. Lymphatic ECs (green panel): After mesoderm specification, EBs are cultured in suspension, co-cultured with OP9 stromal cells, or cultured in feeder-free conditions. The addition of BMP-4, VEGF-A, VEGF-C, angiopoietin-1 (Ang-1), and EGF activates the ERK/Sox18/Prox1 pathways, driving lymphatic differentiation. Key markers, including Prox1, Lyve1, VEGFR3, and Podoplanin, are upregulated by day 7, with mature lymphatic cells obtained by day 30. BMP4: bone morphogenetic protein 4; FGF2: fibroblast growth factor 2; cAMP: 8-bromoadenosine 3′,5′-cyclic adenosine monophosphate; VEGF: vascular endothelial growth factor; COUP-TFII: Chicken ovalbumin upstream promoter-transcription factor II; EphB2: Ephrin type-B receptor 2; EphB4: Ephrin receptor B4; Lyve1: lymphatic vessel endothelial hyaluronan receptor 1; Prox1: Prospero homeobox 1; VEGFR3: vascular endothelial growth factor receptor 3; ERK: extracellular signal-regulated kinase; and Sox18: SRY-Box transcription factor 18.

**Figure 3 ijms-25-13275-f003:**
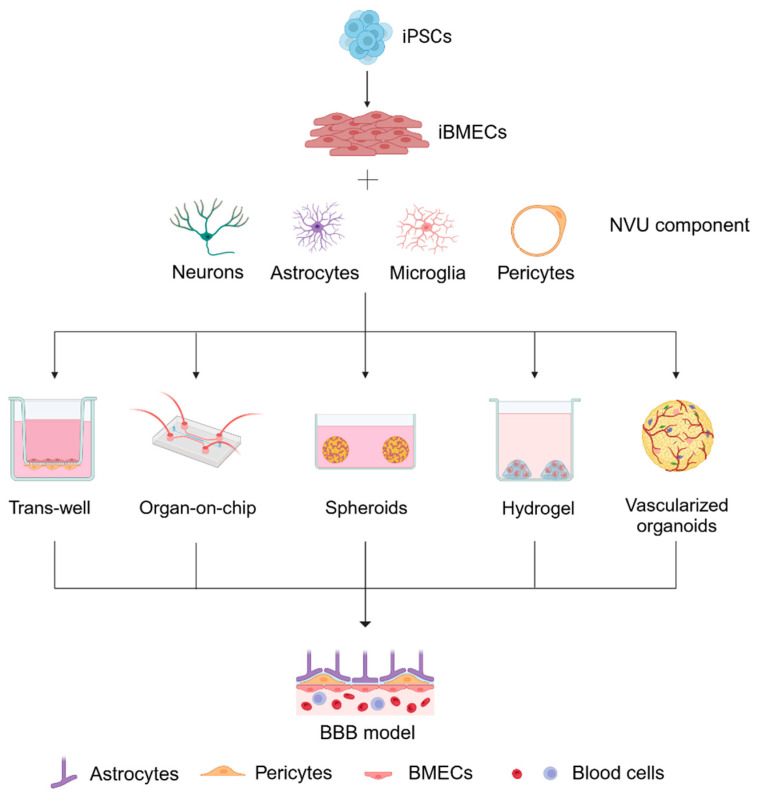
Modeling the blood–brain barrier (BBB) with iPSC-derived brain microvascular endothelial cells (iBMECs). Induced pluripotent stem cells (iPSCs) are differentiated into brain microvascular endothelial cells (iBMECs) and combined with key components of the neurovascular unit (NVU), including neurons, astrocytes, microglia, and pericytes. These cellular elements are assembled in several BBB models, including transwell systems, organ-on-chip platforms, spheroids, hydrogel-based cultures, and vascularized organoids, to mimic the structure and function of the human BBB. Each model offers unique advantages for studying the interactions within the NVU and assessing BBB integrity, permeability, and response to external factors.

**Table 1 ijms-25-13275-t001:** Comparison of primary endothelial cells and iPSC-derived endothelial cells across key parameters, including costs, benefits, limitations, replicative potential, and relevance in disease studies. This table highlights the strengths and challenges of each cell type, emphasizing their applicability to vascular research and therapeutic development.

Parameter	Primary Endothelial Cells (ECs)	iPSC-Derived Endothelial Cells (iPSC-ECs)
Costs	-Initial Costs: High, as ECs are isolated from human tissues, requiring specialized expertise. -Long-term Costs: Limited scalability means recurring expenses for donor-derived samples [58,59].	-Initial Costs: High due to the reprogramming of somatic cells into iPSCs and subsequent differentiation protocols. -Long-term Costs: More economical over time as iPSC lines are renewable and scalable in large quantities [60,61,62].
Benefits	-Physiological Relevance: Closely mimic in vivo ECs, making them ideal for studying native vascular biology and pathophysiology. -Functional Specialization: Can be isolated from specific tissues (e.g., arterial, venous, lymphatic) for targeted studies. -Validation: Widely used with established protocols for functional assays such as angiogenesis, migration, permeability, and adhesion studies [58,59].	-Unlimited Supply: Can be produced in virtually unlimited quantities once the iPSC line is established. -Customization: iPSCs can be genetically modified, enabling the study of specific gene functions and disease phenotypes. -Disease Modeling: Excellent for modeling genetic diseases and exploring rare conditions that cannot be easily studied with primary ECs [60,61,63].
Limits	-Replicative Potential: ECs have a limited number of replications and can undergo senescence after a certain amount of passages. This limits their use in long-term studies. -Isolation Protocol Challenges: Difficult to isolate sufficient cells from small or compromised tissues, such as diseased or elderly donors. -Donor Availability: Difficult to find patients available to donate the cells [58,59,64,65].	-Immature Phenotype: Often exhibit immature or heterogeneous endothelial characteristics compared to primary ECs. -Validation Requirements: Extensive testing is needed to confirm functionality and phenotypic stability, particularly in vivo. -Cost and Time: Differentiation protocols are lengthy and require optimization, which may limit widespread adoption for some studies [61,66].
Replicative Potential	ECs undergo a limited number of cell divisions before senescence, which restricts their use in long-term experiments or large-scale production [58].	iPSC-derived ECs have a theoretically unlimited replicative capacity, allowing for continuous production and repeated experiments [61].
Relevance in Disease Studies	-Strengths: Well-suited for studying diseases that directly affect native endothelium, such as atherosclerosis, hypertension, and diabetes. -Limitations: Limited scalability and donor availability hinder their use in high-throughput drug screening or rare disease modeling [59,67].	-Strengths: Highly relevant for personalized medicine, genetic disease modeling (e.g., familial vascular diseases), and regenerative therapy. -Limitations: Immature or inconsistent phenotypes may reduce their relevance for studying highly specialized vascular conditions [60,61,63].
Ethical Concerns	Limited by donor availability [58,59].	iPSCs can be derived from non-invasive samples such as skin or blood [60].

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
