# Peer review of "Bridging the Gap: Endothelial Dysfunction and the Role of iPSC-Derived Endothelial Cells in Disease Modeling"

_ijms, 2024, doi:10.3390/ijms252413275_

Round 1
Reviewer 1 Report
Comments and Suggestions for Authors
General Comments:
The manuscript is a thorough review of the role of endothelial dysfunction and the potential of induced pluripotent stem cell (iPSC)-derived endothelial cells (iECs) in disease modeling. It is well-organized, with a logical flow of ideas from the background to specific applications. The authors highlight recent advancements in iEC technology and its use in modeling diseases such as cardiovascular, diabetic, and neurodegenerative disorders.
Specific Comments:
1. Abstract:
The future directions mentioned are vague; more specific suggestions are recommended.
Specify key future directions, such as specific technological improvements or applications in drug screening.
2. Introduction:
The introduction is verbose, with some repetition. The connection between endothelial dysfunction and iEC applications could be made earlier to set the stage for subsequent sections.
3. Sections on Mechanisms of Endothelial Dysfunction (2.1, 2.2, 2.3):
Focus on summarizing key pathways and mechanisms, and refer readers to primary sources for intricate details.
4. iPSC-Derived Endothelial Cells:
Discussion of iEC differentiation is fragmented and could benefit from clearer organization.
5. Applications of iECs in Disease Modeling (Cardiovascular, Diabetes, Neurodegenerative Diseases):
Some examples, such as cadmium-induced atherosclerosis, are not representative of broad applications and could be replaced with more universally relevant studies.
The section on neurodegenerative diseases lacks sufficient connection to translational research.
Discuss translational implications for each disease.
6. Figures and Tables:
A summary table comparing iECs with primary ECs across key metrics (e.g., replicative potential, disease relevance etc..) would be very useful
7. Conclusions and Future Directions:
Future directions are generic and lack specific insights. Address current technical and ethical challenges in more detail, such as issues with scalability, reproducibility, and cost.
Minor
Ensure consistency in the use of technical terms, such as consistently referring to induced pluripotent stem cell-derived endothelial cells as "iECs."
Author Response
Answer to Reviewer 1
The manuscript is a thorough review of the role of endothelial dysfunction and the potential of induced pluripotent stem cell (iPSC)-derived endothelial cells (iECs) in disease modeling. It is well-organized, with a logical flow of ideas from the background to specific applications. The authors highlight recent advancements in iEC technology and its use in modeling diseases such as cardiovascular, diabetic, and neurodegenerative disorders.
We thank the reviewer for the comments. We modified and improved the manuscript accordingly. Finally, we would like to thank the reviewer for the comments received that helped us to improve the quality of our manuscript.
- Abstract: The future directions mentioned are vague; more specific suggestions are Specify key future directions, such as specific technological improvements or applications in drug screening.
We entirely revised the abstract and we added a sentence about the future directions on the use of iECs for drug screening and development for the treatment of endothelial dysfunction.
- Introduction:
The introduction is verbose, with some repetition. The connection between endothelial dysfunction and iEC applications could be made earlier to set the stage for subsequent sections.
We revised the introduction according to reviewer suggestions. We rewrote the section, focusing on endothelial dysfunction and iECs applications trying to set the stage for the following sections, as suggested.
- Sections on Mechanisms of Endothelial Dysfunction (2.1, 2, 2.3): Focus on summarizing key pathways and mechanisms and refer readers to primary sources for intricate details.
As suggested by the reviewer we revised all the endothelial dysfunction section. In each section (2.1, 2.2, 2.3) we explained the key mechanisms that cause endothelial dysfunction, and we reorganized the sections to make it clearer and more concise.
- iPSC-Derived Endothelial Cells: Discussion of iEC differentiation is fragmented and could benefit from clearer
We revised the section of ECs differentiation reorganizing it as suggested by the reviewer. We reported the main protocols used to differentiate iPSCs into arteriosus, vascular or lymphatic ECs and we modified Figure 2. In the figure we reported a timeline for the differentiation protocols used for each type of ECs (arteriosus, vascular or lymphatic).
- Applications of iECs in Disease Modeling (Cardiovascular, Diabetes, Neurodegenerative Diseases): Some examples, such as cadmium-induced atherosclerosis, are not representative of broad applications and could be replaced with more universally relevant The section on neurodegenerative diseases lacks sufficient connection to translational research. Discuss translational implications for each disease.
As suggested by the reviewer, we added more relevant and representative studies in each section (Cardiovascular disease, Diabetes and Neurodegenerative disease) and increase the correlation between the cited diseases and the translational applications. In the “cardiovascular disease” section we added a paragraph about the application of 3D models to study endothelial dysfunction. In the “diabetes” section we added some studies on the effects of epigenetic changes on the glycemic memory of ECs. Finally, in the neurodegenerative part we better discussed the connection of the proposed studies and the possible applications in translational research.
- Figures and Tables:
A summary table comparing iECs with primary ECs across key metrics (e.g., replicative potential, disease relevance etc..) would be very useful
We generated a table (Table 1) comparing iECs and primary ECs, as suggested by the reviewer. We use key metrics such as costs, benefits, limits, replicative potential, relevance in disease studies and ethical concerns.
- Conclusions and Future Directions: Future directions are generic and lack specific Address current technical and ethical challenges in more detail, such as issues with scalability, reproducibility, and cost.
The conclusion section was revised according to reviewer comments. We added several sentences about the future research that can be developed using iECs to study endothelial dysfunction. Moreover, we focused on the ethical and technical challenges in the use of iECs. We addressed these issues also in the Table 1.
Minor: Ensure consistency in the use of technical terms, such as consistently referring to induced pluripotent stem cell-derived endothelial cells as "iECs."
We ensure consistency in the use of technical abbreviations through the manuscript.
Reviewer 2 Report
Comments and Suggestions for Authors
Please see attached.

Acceptable with minor revisions. Please see my comments to the authors for further details.
Author Response
Answer to Reviewer 2
In the manuscript (ijms-3350675) “Bridging the Gap: Endothelial Dysfunction and the Role of iPSC- Derived Endothelial Cells in Disease Modeling,” the authors reviewed the in-vitro models of endothelial dysfunction, emphasizing the advantages of induced pluripotent stem cell (iPSC) derived endothelial cells (iECs) as a disease model. My comments/suggestions on each section of the manuscript are as follows:
- The authors mentioned in the abstract that “primary ECs show some limitations” (line 15). I recommend that they briefly outline these limitations to enhance the readability of the abstract.
As suggested by the reviewer we added the limitations of primary endothelial cells.
- The “Introduction” is brief and mainly focuses on the functions of endothelial cells (ECs), but it does not provide sufficient background for the overall manuscript. Additionally, the meaning of the latter part of the following sentence (underlined) is unclear. “Considering the key role of ECs in the maintenance of the vessel's stability, the ECs dysfunction is strictly connected to several vascular pathologies such as coronary artery disease, hypertension and diabetes showing dysfunctional behavior compared to healthy ECs”.
We thank the reviewer for the comment. We revised the section, focusing on endothelial dysfunction and iECs applications in cardiovascular disease, diabetes and neurodegenerative disease trying to set the stage for the following sections, as suggested. Additionally, the sentence suggested by the reviewer was deleted, since all the introduction was revised.
- Section 2 “Endothelial Dysfunction” discusses the influence of oxidative stress, shear stress, and extracellular matrix organization on endothelial function and dysfunction. This section contains several linguistic ambiguities. A few examples are provided below.
- In lines 55-56, the authors stated that “Various risk factors contribute to the onset of endothelial dysfunction, making it a multifactorial phenomenon”. Please briefly list the risk factors here or indicate if the risk factors are discussed below.
We revised the sentence and the section of endothelial dysfunction according to reviewer comment.
The meaning of the following phrases is unclear. “under physiological conditions” (lines 75-76) or “under physiological flow” (line 119). What does physiological condition or flow mean? Is it a normal or an abnormal condition? “When disturbed” (line 119) or “in disturbed flow conditions”. It is not clear what disturbed flow means in this context.
We revised the phrases as suggested and we modified the sentences containing “physiological, abnormal or disturbed flow” with low or oscillatory stress referring to low (time-average: 10–12 dyne/cm 2) or oscillatory shear stress (time-average: close to 0).
- Lines 119-121: Does endothelial dysfunction lead to diabetes? Please revise.
We modified the sentence since we revised the whole paragraph.
- Section 3 “Primary Endothelial Cells and Their Limitations” reviewed the examples of human umbilical vein endothelial cells (HUVECs) and mononuclear cell-derived early and late endothelial progenitor cells (EPCs) as primary cell models to investigate endothelial dysfunction and disease.
- The outlined limitations are generic.
- There are also a few grammatical errors:
“ECFCs have been used for several applications as diabetes” (lines 216) “hematological diseases, as hemophilia A” (lines 223) “the regulation of ECM-related genes as nidogen 2” (lines 232)
We better discussed the limitations of the use of primary ECs and for clarification we added a table (Table 1) reporting costs, benefits, limits, replicative potential, relevance in disease studies and ethical concerns. Finally, we corrected the grammatical errors as suggested.
- Section 4 “iPSC-Derived Endothelial Cells”.
- Including a detailed protocol for differentiating iPSCs into endothelial cells would enhance this section and benefit readers.
We revised the section of ECs differentiation reorganizing it. We reported detailed protocols to differentiate iPSCs into arteriosus, vascular or lymphatic ECs and we modified Figure 2. In the figure we reported a timeline for the differentiation methods used for each type of ECs (arteriosus, vascular or lymphatic).
- Section 5 “iECs Applications for the Study of the Endothelial Dysfunction in Pathological Conditions”
- A typographical error in the heading: “Pathological”.
- Subsection 5.1: Expanding the discussion on novel 3D models, including microfluidic systems and organoids, will enhance the disease-modeling potential of iPSC-derived-ECs for readers.
As suggested, we expanded the discussion about the use of 3D models from iECs to enhance the understanding for the reader.
- Section 5.2: The study by Gorashi et al., referenced by the authors in this section, indicated that iPSC-derived ECs derived from diabetic patients showed diabetes- related pathologies. Were these ECs cultured in a diabetic medium? If not, is there any information regarding the epigenetic memory retained by the reprogrammed iPSCs/iECs? Generally, iPSC reprogramming alters most of an individual's epigenetic memory in the reprogrammed cells, including any diabetes-associated changes except for the genetic predisposition. A discussion on this issue could benefit readers.
In the Gorashi et al. study the ECs were cultured in high glucose medium to mimic diabetic environment. Moreover, we added a couple of sentences about the effects of epigenetic changes on the glycemic memory of ECs.
- Including a section or at least a paragraph on the limitations of iPSC-derived iEC models will offer readers a more balanced perspective and enhance their understanding.
As suggested by the reviewer, we included a table (Table 1) showing the limitations of iECs to make it more concise and clearer for the readers.
Finally, we thank the reviewer for the comments. We modified and improved the manuscript accordingly. Finally, we would like to thank the reviewer for the comments received that helped us to improve the quality of our manuscript.